# Influence of Pre-Ionized Plasma on the Dynamics of a Tin Laser-Triggered Discharge-Plasma

**Qiang Xu [1], Xiaolong Deng [1,\*], He Tian [1], Yongpeng Zhao [2] and Qi Wang [2]**

[1] College of Science, Northeast Forestry University, Harbin 150040, China; nefu_xq@nefu.edu.cn (Q.X.); tianhe@nefu.edu.cn (H.T.)
[2] National Key Laboratory of Science and Technology on Tunable Laser, Harbin Institute of Technology, Harbin 150080, China; zhaoyp3@hit.edu.cn (Y.Z.); qiwang@hit.edu.cn (Q.W.)
[\*] Correspondence: 469562079@nefu.edu.cn

**Abstract:** The effect of laser-current delay on extreme ultraviolet emission by laser-triggered discharge-plasma has been investigated. Typical waveforms for current, voltage, laser signals, and X-ray signals have been compared. Theoretical tin spectra were simulated among the electron temperature ranges from 10 to 50 eV to compare with the experimental results. The results show that longer laser-current delay time is propitious to increase the steady-state time of plasma at high temperatures, and it increases the intensity and spectral purity of 13.5 nm emission in 2% band. The 13.5 nm radiation intensity increases about 120% with the delay increasing from 0.7 to 5 μs, and the extreme ultraviolet (EUV) emission conversion efficiency (CE) increases from 0.5% to 1.1%.

**Keywords:** EUV emission; laser-triggered discharge plasma; pre-ionized plasma; tin spectra; 13.5 nm emission

## 1. Introduction

In order to keep pace with Moore's law, photolithography has been widely used in the semiconductor industry during the last few decades. At present, the 13.5 nm extreme ultraviolet lithography (EUVL) is one of the photolithography methods with the most potential [1]. The wavelength is 13.5 nm emission in a 2% bandwidth, which is mainly for the molybdenum-silicon (Mo/Si) multilayer mirrors [2] and the photo resistance [3]. Due to the high convention efficiency of 13.5 nm emission, Sn has been widely used as an extreme ultraviolet (EUV) source [4]. Two types of EUV sources are used for EUVL, which are laser-produced plasma (LPP) [5,6] and laser-triggered discharge plasma (LTD) [7]. The LTD source has the features of being a low-cost system, high repetition rate, selective spectral wavelength according to target, potential high repetition-rate operation, and precise timing controllability of a light pulse. In this way, it can be used not only for EUVL, but also for EUV-induced plasma [8], nano-patterning [9], and X-ray microscopy [10,11]. Moreover, it can be used to inspect the key components of the EUVL system, such as the mask and so on [12].

In recent years, a number of experiments have been conducted with LTD sources. The dynamics of the Z-pinch plasma are investigated by different methods [13,14], which show that the electron density is about $10^{18}$–$10^{19}$ cm$^{-3}$, and the electron temperature is about 20–40 eV. With a special three electrodes system, Lu et al. investigated the post-discharge stage of the LTD EUV source [15], and it shows that that electrical recovery lasts for a few tens to several hundreds of microseconds, and the repetition of the source can exceed 70 kHz, which is useful to achieve high power of the EUV source. Parameters and types of lasers are also investigated by V. M. Borisov et al. and G. A. Beyene et al. The former reports that the maximum conversion efficiency is almost the same for $CO_2$, Nd:YAG, XeF, and KrF laser, and the dependence of the EUV source efficiency on the laser power density is

obtained as well [16]. The latter studies the EUV spectra generated by picosecond and nanosecond laser triggering under different laser energy. It reports that ps-triggering is better in order to get a higher EUV conversion efficiency and narrower spectral profiles [7]. Moreover, the effect of current rise time, electrical energy, and inter-electrode distance are also reported [16,17].

The plasma characteristics during the Z-pinch and recovery stage have been studied extensively by using optical techniques or spectroscopy. Until now, the research on the expanding of the pre-ionized plasma for Sn LTD sources has been reported very little. For the LTD source, the trigger laser generates an expanding pre-ionized plasma, which is the discharge medium for the latter process. The condition of the pre-ionized plasma varies with the expanding, and it will dictate the subsequent compression dynamics and the spectral properties of the Z-pinch. Moreover, the pre-ionized plasma results in a much lower threshold for the breakdown voltage as well [18]. In this way, study of the delay between the triggering laser and the current, may be useful for the optimizing of the EUV emission. In this paper, we focus on the dynamics under different laser-current delays by time-resolved EUV spectroscopy.

## 2. Experimental Apparatus

An experimental schematic diagram of tin LTD is given in Figure 1, which is mentioned in the previous works [19]. In all experiments, a 3 mm thick and 60 mm diameter high purity, planar tin target was the source for EUV generation. An Nd:YAG laser beam, produced by Continuum Minillite II laser system (8 ns pulse width, 1–50 mJ @1064 nm, 1–10 Hz), having a pulse energy of 20 mJ, was focused by an $f$ = 10 cm focal length planoconvex BK7 glass lens, which was placed in the center of the hollow cathode. The laser beam was injected perpendicularly onto the tin target. The power density was $10^{10}$ W/cm$^2$ at a focal spot diameter of 45 μm FWHM. The distance between the hollow tungsten-copper anode and the tin cathode was 5 mm. Moreover, the laser pulse, which was used to produce a pre-ionized plasma, was not high enough to produce a detectable EUV emission. The tin cathode was rotated after five laser shots to avoid the effect of the electrode ablation [20]. In this paper, the measured spectrum is the average of the five shots.

A 1 m Rowland circle grazing incidence spectrometer (GIS, McPherson 248/310 G) and an X-ray CCD (Andor, DO920P-BN) were used to record the time-integrated spectra. The spectral resolution is better than 0.04 nm. The spectrometer axis was vertical to the axis between anode and cathode electrodes, and coincided with the focused laser. An EUV energy monitor E-Mon (Bruker Advanced Supercon GmbH, 00M-EM-001), consisting of a Zr filter, two Mo/Si multilayer mirrors, and an X-ray photo-diode was used to measure pulsed EUV-energies and calibrated the GIS spectra detection system at central wavelength of 13.5 nm with a bandwidth of 2%.

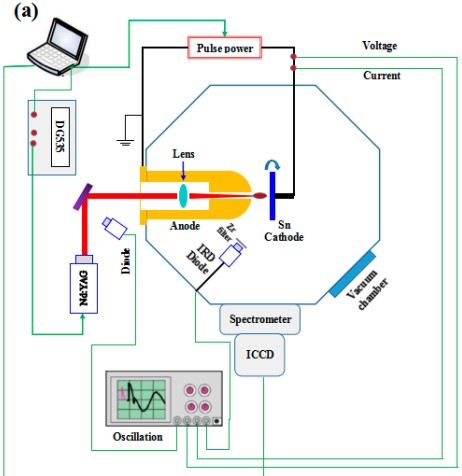 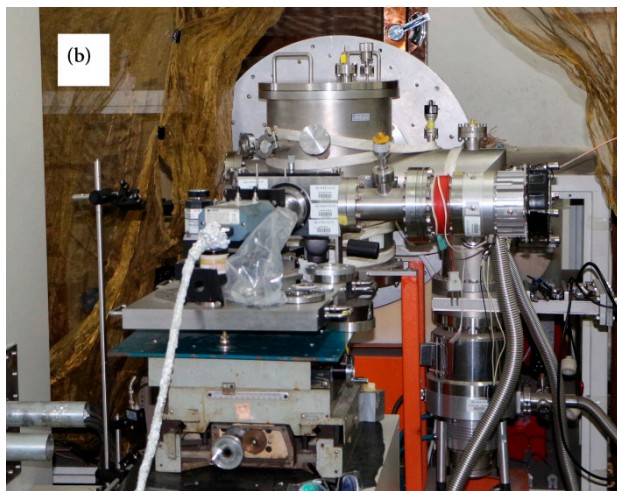

**Figure 1.** *Cont.*

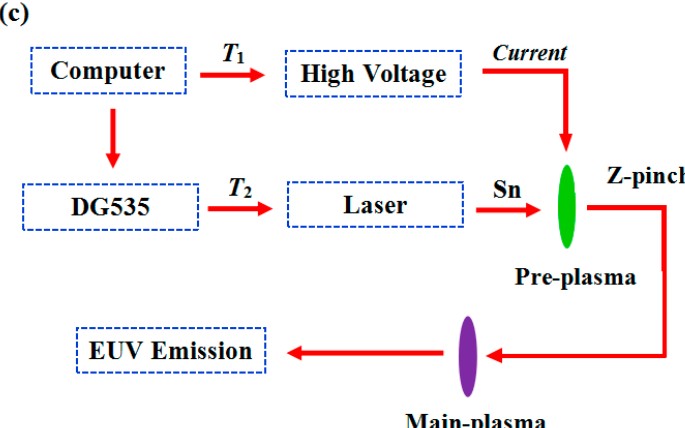

**Figure 1.** Schematic of the experimental setup used for a laser-triggered discharge produced in tin plasma. (**a**) Schematic of the whole setup. (**b**) Photograph of the experimental setup from the side of the grazing incidence spectrometer of the vacuum chamber. (**c**) Time series diagram of the experimental setup.

In order to avoid self-discharge and absorption of EUV emission, the vacuum chamber pressure was typically in the order of $10^{-3}$ Pa, maintained by a mechanical pump and a turbo molecular pump. A Rogowski coil (Tektronix P6015A) and a high voltage probe (Pearson 5046) were used to measure the current and the voltage respectively. A diode (Thorlabs, DET10A/M) was used to monitor the laser signal. An EUV photo-diode (IRD, AXUV20HS1BNC) was used to observe the X-ray signal from a discharge. A 500 nm thick zirconium metal filter was placed in front of the IRD diode to select wavelengths from 6.5 nm to 16.8 nm [21]. These outputs of the current, the voltage, the signal of the laser, and the X-ray signal in one shot were recorded on a digital oscilloscope (Tektronix, MDO3054).

The power supply was based on three-level magnetic pulse compression, which differs from conventional LTD sources [17,22]. The high voltage can be generated and maintained by the pulse power system. When the pre-ionized plasma bridges the anode and the tin cathode, the current is generated. Then, the pre-ionized plasma is pinched to the center by the Lorentz force. In this way, high-temperature plasma is made to radiate EUV emissions. By modifying the delay between laser and voltage, this power supply can command the discharge process and the condition of the plasma. Figure 1b shows the time series diagram of the experimental setup. A computer gives two signals simultaneously to trigger the power supply and the digital delay generator (SRS, DG535). The inherent delay $T_1$ is defined by the pulse power system and the final Z-pinch device, and it is a constant. Meanwhile, the delay $T_2$ controlled by the DG535 was used to adjust the delay between the current and the laser signal.

Figure 2 shows two typical waveforms for the voltage, the current, the IRD diode X-ray signal, and the laser signal under the laser-current delay d$t$ = 0.68 μs and 1.60 μs respectively. The laser-current delay d$t$ (~10% of laser pulse to 10% of current) is defined as the time interval between the laser irradiation (pre-ionized plasma) and the onset of the current (breakdown). The increasing time and full width at half maximum (FWMH) of the current are 100 ns and 260 ns, respectively, and the typical current and voltage values are about 8 kA and 20 kV, respectively. The capacitance of the last loop is 120 nF, which delivers 24 J energy to the pre-ionized plasma. According to the experiments, no discharge will occur at d$t$ smaller than 0.68 μs, which is mainly determined by the electrical gap distance, the laser irradiation, and the target material [23]. In fact, many experiments have been done by increasing $T_2$, which is shown in Figure 1b. When $T_2$ is large enough, the pre-ionized plasma will be generated after that the high voltage is loaded between the two electrodes. In this way, the high voltage will be maintained, which is similar to Figure 2A. The time interval was defined as being between the falling edge of the voltage and the onset of the current as d$t'$, which is shown in Figure 2A. The experimental results show that d$t'$ is increasing with the increasing of $T_2$. However,

the laser-current delay dt (~10% of laser pulse to 10% of current) is almost 0.68 μs under different $T_2$. There are two peaks of the X-ray signals. The first peak is generated by the collision between the anode and the electrons, which is generated by the laser and accelerated by the electric field [22]. The second peak, which is corresponding to the dip in the current signal, is due to the Z-pinch formation by the current [14,22]. By comparing the two X-ray signals, it can be found that there are much more smaller peaks for the second peak when d*t* is larger. Moreover, the pinch times, which are defined as the onset of the current and the dip of the current, are 116.4 ns and 105.2 ns for condition (A) and (B) respectively. Due to the expanding of the pre-ionized plasma, lower density of ions will be obtained when the laser-current delay is larger. In this way, the pre-ionized plasma will be pinched much more drastically and faster according to the Bennett equilibrium and the snow-plow model [24,25]. Multiple pinches may occur when d*t* is larger as well [26]. Moreover, the FWHM of the X-ray signal is larger under larger d*t*, according to Figure 2.

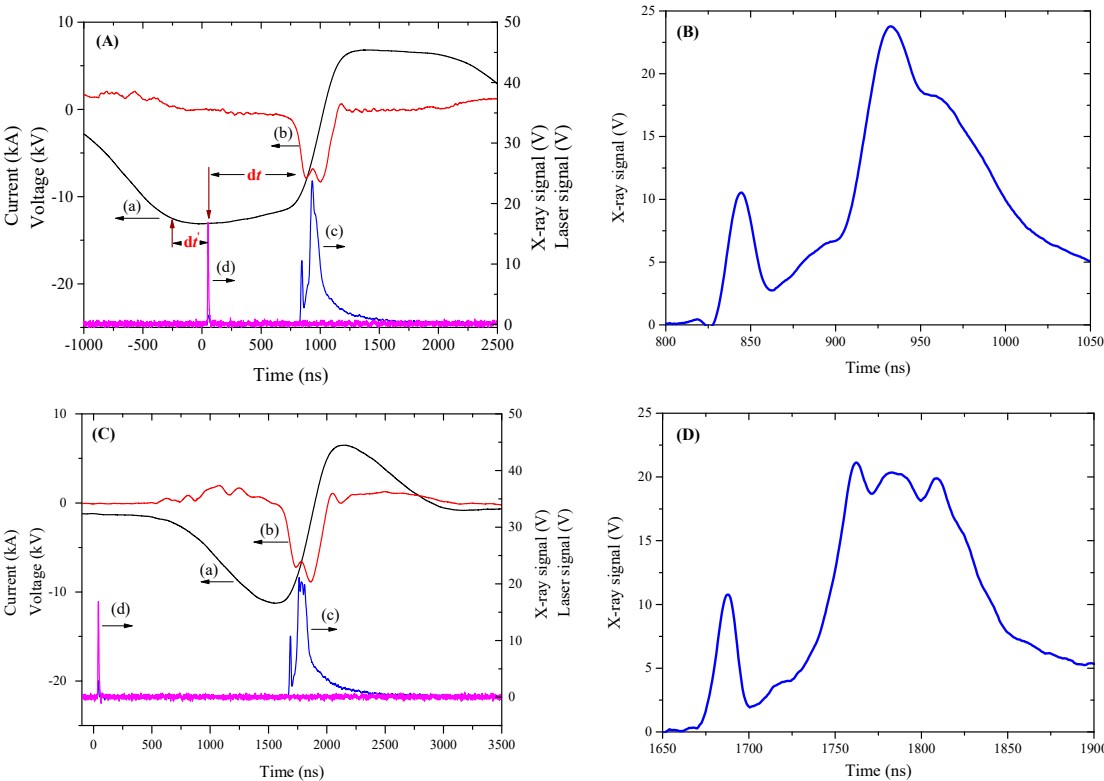

**Figure 2.** Typical waveforms for the voltage (a, black line), current (b, red line), IRD diode X-ray signal (c, blue line), and the laser signal (d, pink line) under two laser-current delays. (**A**) The high voltage was maintained until the pre-ionized plasma was ignited, and the electrical breaks under the laser-current delay 0.68 μs. (**C**) The tin pre-ionized plasma was generated by laser long before the high voltage was loaded between the anode and the tin cathode. (**B,D**) The X-ray signals, which are enlarged by 20 times, have been magnified corresponding to conditions (**A**) and (**C**) respectively.

## 3. Theoretical Calculations

To simulate the emission from tin ions around 13.5 nm, the transition probability (*gA*) and oscillator strengths (*gf*) versus wavelength (*λ*) are calculated for a range of tin ion stages by Hartree–Fock configuration interaction (HFCI) Cowan codes [27]. Here configuration interaction (CI) is taken into account between the $4d^n$, $4d^{n-1}mf$, $4d^{n-1}tp$, and $4p^5 4d^{n+1}$ (*m* = 4–6, *t* = 5–7, *n* = 1–8) configurations, as it will considerably influence the *gA*, the wavelength [28], and narrow the bandwidth. The Slater Condon parameters are set according to Gerry O'Sullivan [29]. There are tens of thousands of transition lines in the EUV band for highly charged tin ions, and the lines from different charge states and

different transition arrays overlap with each other, forming an unresolved transition array (UTA) or quasi-continuous band. Therefore, it is necessary to broaden the spectral lines when simulating the spectral lines. Statistical methods are used to calculate the averaged wavelength versus ion stages according to Equation (1) [30]. As shown in Figure 3A, the 13.5 nm emission in 2% bandwidth (13.365–13.635 nm) is mainly generated by the $Sn^{8+}$–$Sn^{13+}$ 4d–4f and 4p–4d transitions.

$$\overline{\lambda} = \sum_{i=1}^{N} (\lambda_i \cdot gA_i) / \sum_{i=1}^{N} gA_i \quad \Delta\lambda = \sqrt{\sum_{i=1}^{N} [(\lambda_i - \overline{\lambda})^2 \cdot gA_i] / \sum_{i=1}^{N} gA_i} \tag{1}$$

**Figure 3.** Theoretical results of averaged wavelength and ion fraction. (**A**). Average wavelength verse ion stage for tin for the transitions 4d–5p (black, square), 4d–6p (red, circle), 4d–4f (blue, up triangle), 4d–5f (pink, star), and 4p–4d (green, diamond). (**B**). Fractional ion stage distributions and average ions stage in an LTD tin plasma vs $T_e$ at $n_e = 10^{19}$ cm$^{-3}$ evaluated using the Colombant and Tonon's collisional-radiative model. Ion stages which can emit 13.5 nm emission are shown in red.

A collision–radiation (CR) model of Colombant and Tonon [31] is used to approximately estimate the ion stage distribution. The ion fraction $f_z$ of charge $z$ is given in Equation (2) for rate coefficients of collisional ionization $S$, radiative recombination $\alpha_r$ and three-body recombination $n_e\alpha_{3b}$. Figure 3B shows the ion fractions and average ions stages versus electron temperature for tin to the 15th ion stage during the electron temperature range from 1 to 70 eV. The electron density is choosing as $10^{19}$ cm$^{-3}$, which is typical for LTD source [13,14]. Moreover, the electron density has almost no effect on the ion stage distributions when the electron density is under $10^{21}$ cm$^{-3}$ [32]. It is seen that the optimal temperature for $Sn^{8+}$~$Sn^{13+}$ ions, which can emit the emission around 13.5 nm, mainly lies in the range of 15–50 eV. The dominant fractional abundance of $Sn^{8+}$~$Sn^{13+}$ ions should be less than 60% of the total ions for a given $T_e$.

$$f_z = \frac{n_{z+1}}{n_z} = \frac{S(z, T_e)}{\alpha_r(z+1, T_e) + n_e\alpha_{3b}(z+1, T_e)} \tag{2}$$

The LTD plasma is optically thin when comparing it with the LPP plasma [7]. In this way, the opacity effect is ignored. A Boltzmann distribution appropriate to the temperature is used to weight the population of the upper states for each stage [33]. Figure 4 shows the simulated tin spectra as a function of temperature and wavelength. It is clear from these simulated spectra that the optimum electron temperature is around 30 eV, where the emission around 13.5 nm is highest. The spectra in the range 14–16 nm, which are mainly generated by the 4d–df and 4p–4d transitions of $Sn^{6+}$–$Sn^{8+}$ ions, can be seen when the electron temperature is lower than 20 eV. Moreover, with the increasing of $T_e$, the UTA bandwidth is getting narrower as well.

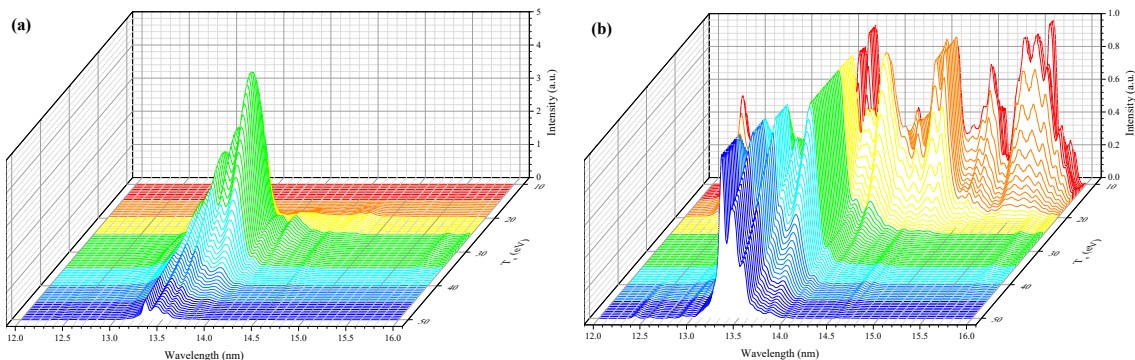

**Figure 4.** Theoretical tin spectra as a function of temperature and wavelength, convolution with a Gaussian profile of 0.06 nm full width at half maximum. (**a**) The variation of absolute emission as a function of $T_e$. (**b**) Normalized to the peak at each temperature to compare the variation in profile.

## 4. Experimental Results and Discussion

Time-integrated tin EUV emissions are presented in Figure 5 under different laser-current delays. The energy and focal-spot diameter of the laser is fixed. Moreover, the discharge current is basically unchanged under different delays when the voltage is maintained. In this way, the input electrical energy is almost the same. Since the experimental spectrum is time-integrated over the laser pulse duration, it is not possible to characterize the emission by a single electron temperature over a range of power densities and electron temperatures. Figure 6 shows the comparisons with the theoretical simulations for steady-state electron temperatures 32, 20, and 12 eV, respectively. The fractions of the contributing ion stages for each electron temperature are given in the inset. It can be found that the theoretical result from the temperature of 32 eV fits the short-wavelength spectra well, especially between 13.0 and 13.5 nm. The long-wavelength spectra are in good agreement with those of the temperature of 12 eV when the delay is 1.5 µs. In this way, the electron temperature of the plasma should be in the range of 12 and 32 eV during the Z-pinch process. According to Figures 3A and 6a, the spectra between 13 and 14 nm mainly originate from $Sn^{10+}$ and $Sn^{12+}$, and these ions mainly exist under the electron temperature 32 eV. In this way, the experimental data and the simulation data can agree well. The condition for Figure 6c is for the same reason. However, it can be seen from Figure 3A that the origin of spectra in the range of 14–15 nm is particularly complex, which are mainly coming from $Sn^{6+} \sim Sn^{13+}$ ions. These ions can exist in a very large temperature range. At the same time, the measured spectrum is a time integral spectrum, which corresponds to the overall situation of plasma radiation spectrum at different temperatures during the whole Z-pinch process. This means that the spectra in this band can not match the fitting spectra at a certain temperature, which is also the main reason why the theoretical and experimental spectra in Figure 6b cannot fit well.

According to Figures 5 and 6a, the intensity between 13 and 14 nm, which mainly originates from $Sn^{10+}$ and $Sn^{12+}$, is increasing with the delay from 0.7 to 5.0 µs. Meanwhile, as shown in Figure 6b,c, there is some emission from the lower ion-stages, and the strongest emission is from $Sn^{6+}$ to $Sn^{9+}$ (14–16 nm). As shown in Figure 5b, the fraction of the emission between 14 nm and 16 nm is going down with the increase of the delay. As mentioned, the experimental spectra are time-integrated, which indicates average charge state during the plasma lifetime. In this way, the charge state increases with the delay, which means that the average electron temperature of the plasma is increasing. As the delay increases, the density of the plasma decreases due to expanding. In this way, the Z-pinch process gets more intense according to the Bennett equilibrium. The electron temperature gets higher as well. Moreover, there is a small peak around 12.5 nm, which is generated by the $Sn^{7+}$ 4d–5f transition. It almost cannot be detected when the delay is larger than 3.0 µs according to Figure 5, which implies that the ratio of the ion stages lower than $Sn^{7+}$ is too little by comparing with higher ion stages.

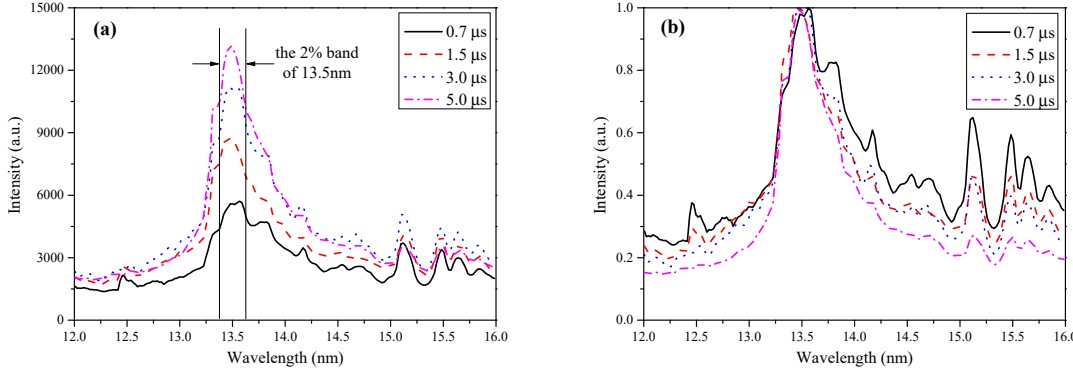

**Figure 5.** Extreme ultraviolet (EUV) spectra and 13.5 nm intensity in 2% band under different laser-current delays. (**a**) Original spectra between 12.0 and 16.0 nm, the spectra marked between the two black lines are the 13.5 nm emission in the 2% band. (**b**) Normalized for each comparison.

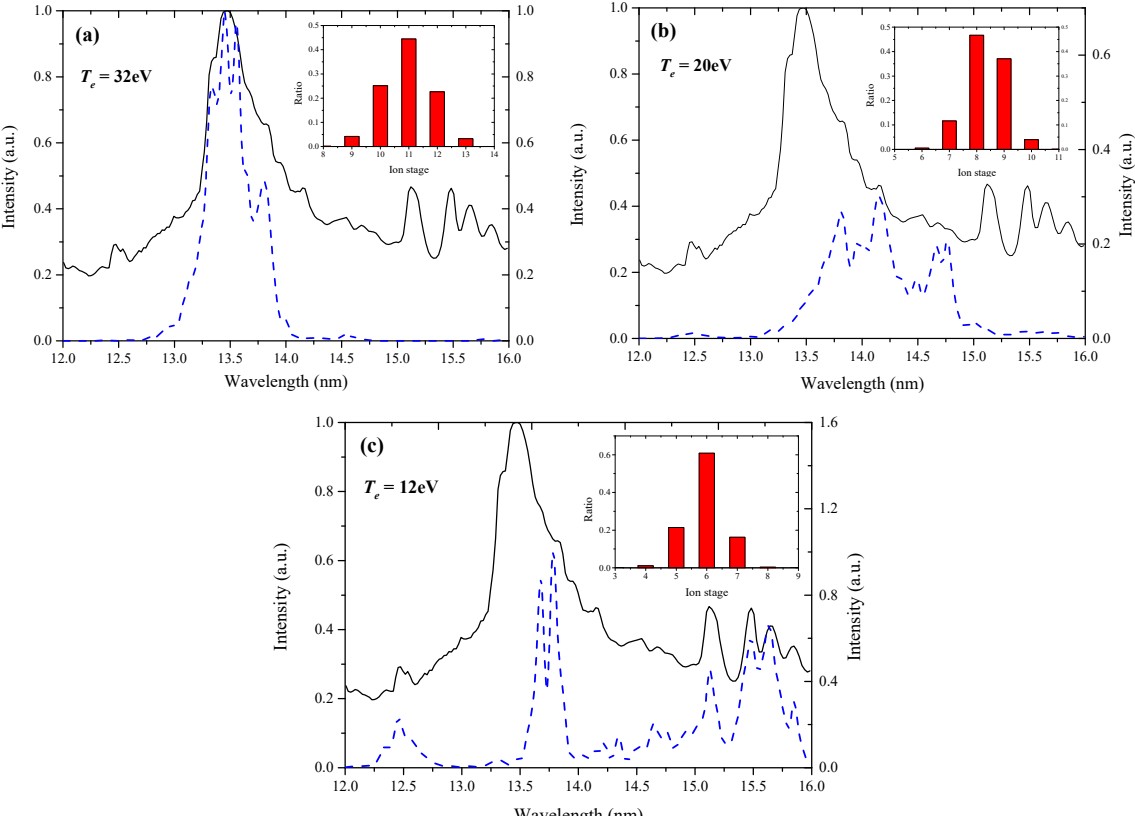

**Figure 6.** Comparison of theoretical spectra (blue, dash line) with the experimental spectra (black, solid line) at electron temperatures 32 eV (**a**), 20 eV (**b**), and 12 eV (**c**) for d*t* of 1.5 μs. The fractions of the contributing ion stages for each electron temperature are given in the inset.

Figure 7 shows the 13.5 nm intensity in the 2% band (red, circle) and spectral purity (blue, star) as functions of laser-current delay. Here the spectral purity is defined as the ratio of 13.5 nm intensity in the 2% band to the range of 12.0 to 16.0 nm. It can be found that the 13.5 nm intensity increases about 120% when the delay increases from 0.7 to 5 μs. The EUV conversion efficiency (CE) is increasing from 0.5% to 1.1%. The EUV CE is determined by the ratio of the energy radiated in a 2% bandwidth around 13.5 nm to the input electrical energy [5]. As shown in Figure 7, the EUV CE is increasing with the laser-current delay, but the increase does not have a linear relationship. In contrast, the stability of the light decreases, which is harmful to the source. Moreover, the 13.5 nm intensity is almost the

same as that of 4 μs when the delay is longer than 4 μs. When the delay is too large, the density of the pre-ionized plasma is too little, and the voltage becomes hard to be broken down as well.

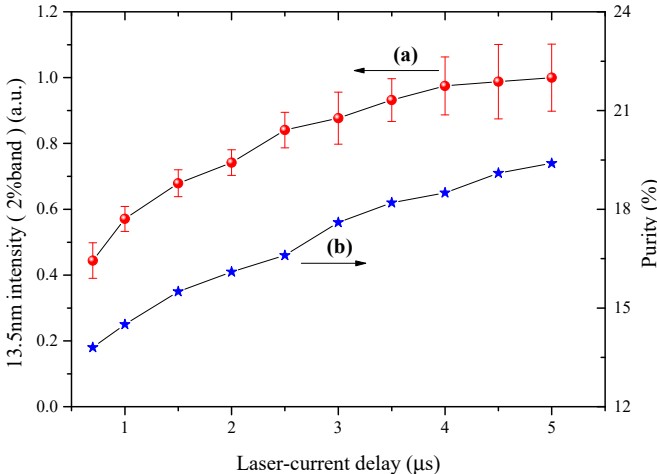

**Figure 7.** Relationships between 13.5 nm intensity in the 2% band (a, red circular) as well as spectral purity (b, blue star) and laser-current delay.

For the EUV source, the 13.5 nm emission intensity in the 2% band is defined by the lifetime and density of the plasma which can generate spectra around 13.5 nm. The plasma lifetime and density are decided by the recombination and ionization of the ions. Hence, we should keep the plasma, which is useful for emitting 13.5 nm emission, as long as possible. According to Figures 5 and 6, it can be found that the mean temperature is increasing with the delay, which means that the plasma is in a high-temperature state for a longer time during the Z-pinch process. In this experiment, the optimal delay for 13.5 nm emission in the 2% band, which is mainly generated by the $Sn^{8+}$–$Sn^{13+}$ $4d$–$4f$ and $4p$–$4d$ transitions, has not been obtained. The density of these ions is decided by the density of the plasma. Moreover, it is also decided by the abundance ratio between the density of $Sn^{8+}$–$Sn^{13+}$ ions and the total density of the plasma. Longer laser-current delay leads to a lower density. On the other hand, the increasing of $Sn^{10+}$ and $Sn^{12+}$ ions' intensity with the delay from 0.7 to 5.0 μs means that the ratio should increase. In other words, a limited current leads to a limited Z-pinch Lorentz force. When the Z-pinch Lorentz force is limited, the pre-ionized plasma cannot be pinched accurately as well. As a result, the electron temperature is not high enough to get enough $Sn^{8+}$–$Sn^{13+}$ ions. In this way, we will try to increase the discharge current to reduce the optimal laser-current delay, and it will increase the EUV CE as well. Moreover, finite laser power will restrict the density of the pre-ionized plasma as well, so we will study the influence of laser-current delay under different laser power by changing the focus distances.

## 5. Summary

In conclusion, we studied the effect of laser-current delay on tin LTD plasma by experiments and theory. Due to the expanding of the pre-ionized plasma, multiple pinches may occur when the delay is larger. The shape of the spectra and 13.5 nm (2% band) intensity varies with the delay. Spectrum simulations predict that the 13.5 nm intensity reaches the maximum when the electron temperature is close to 30 eV, where the ratio of $Sn^{10+}$ ions is highest. With fixed discharge energy but higher laser-current delay, lower vapor velocity leads higher electron temperature and higher intensity 13.5 nm emission in the 2% band, which increases about 120%. The EUV CE increases from 0.5% to 1.1% when the delay increases from 0.7 to 5 μs. The results show stronger dependence of the 13.5 nm intensity on the laser-current delay. By tuning the delay, the bandwidth of the UTA is narrower as well. Moreover, the stability of the 13.5 nm emission decreases as well.

**Author Contributions:** The authors have worked together to complete this research.

**Funding:** We acknowledge funding from the Fundamental Research Funds for the Central Universities (No. 2572018BC18), and Northeast Forestry University Students' Innovation and Entrepreneurship Training Program (No. 201610225100).

**Conflicts of Interest:** The authors declare no conflict of interest.

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
