# Peer review of "Influence of Pre-Ionized Plasma on the Dynamics of a Tin Laser-Triggered Discharge-Plasma"

_applsci, doi:10.3390/app9234981_

Round 1

Reviewer 1 Report

This paper describes the effect of laser-current delay on the spectral purity of tin plasmas produced by a laser-triggered discharge (LTD) source, which is possibly applied to the extreme ultraviolet (EUV) lithography around 13.5 nm. The methodology of this study is almost the same as the authors' recent work on Gd plasmas. The authors measured EUV spectra from LTD plasmas for several different laser-current delay. The measured spectra have been compared with the calculations done by the Cowan code. The result of the comparison implies that the average ion charge would be higher for longer laser-current delay due to higher electron temperature. The spectral purity increases with increasing laser-current delay up to 5.0 us.
Though English should be more improved, the reviewer judges that the manuscript is basically acceptable for publication. The reviewer suggests minor amendments as listed below. Most of them are about English corrections to improve readability:

Page 1, line 10, 39:
"invested" -> "investigated"

Page 1, line 11:
"Theory tin ..." -> "Theoretical tin ..."

Page 1, line 15-16:
The expression "With the EUV CE increases from 0.5% to 1.1%." is not a complete English sentence. Also, "EUV" and "CE" should be spelled out at their first appearance.

Page 1, line 22-23: The authors describe "At present, the most popular photolithography method is 13.5 nm extreme ultraviolet lithography.". However, the reviewer thinks that the EUV lithography be still under development and not the most popular.

Page 1, line 23:
"choosing" -> "chosen"

Page 2, line 61:
"The laser beam is perpendicularly ..." -> "The laser beam is injected perpendicularly ..."

Page 2, line 66:
The sentence "According the experiments, five shots the spectra are not found to change significantly." is hard to understand. English should be improved.

Page 3, Figure 2:
The letters for the insets are too small to read.

Page 3, Figure 2:
The color for the X-ray signal does not match with the text in the caption.

Page 3, line 105:
"show" -> "shows"

Page 3, line 111:
"24J power energy" -> "24J energy"

Page 3, line 111:
The authors describe "As shown in Figure 2(a), no discharge occurs at dt smaller than 0.68 μs, ...". However, Figure 2(a) does not show such a fact.

Page 3, line 116:
"By comparing with the two ..." -> "By comparing the two ..."

Page 4, line 134:
"... used to calculated the averaged wavelength verse ion stages ..." -> "... used to calculate the averaged wavelength versus ion stages ..."

Page 4, Figure 3:
The "(a)" and "(b)" have not been denoted in the figures.

Page 4, Figure 3:
Why the average wavelength of 4p-4d transition for ion stage of 5 is irregular?

Page 4, Figure 3:
"Theory results ..." -> "Theoretical results ..."

Page 4, Figure 3:
The colors for the 4d-5f and 4p-4d do not match with the text in the caption.

Page 4, line 145:
"... in the Equation (2) for collisional ..." -> "... in the Equation (2) for rate coefficients of collisional ..."

Page 5, line 157:
"show" -> "shows"

Page 5, Figure 4:
"Theory tin ..." -> "Theoretical tin ..."

Page 5, Figure 5:
The origin of the vertical axis should start with zero.

Page 6, line 195:
"Figure 6" -> "Figure 5"

Page 6, Figure 6:
The origin of the vertical axis should start with zero.

Page 6, Figure 6:
The color for the theoretical spectra does not match with the text in the caption. Also, the letters for the insets are too small to read.

Page 6, line 204:
"CE" should be spelled out at their first appearance.

Page 6, line 207-208:
"On contrast, ..." -> "In contrast, ..."

Page 6, Figure 7:
The origin of the vertical and horizontal axes should start with zero.

Page 6, line 215:
The expression "..., here the plasma is referred to that which can generate 13.5 nm emission." is hard to understand what it means. It should be reworded.

Page 7, line 222:
What is "the ratio"?

Page 7, line 224:
"... should increases." -> "... should increase."

Page 7, line 225:
The expression "... makes the pre-ionized plasma cannot be pinched ..." is grammatically incorrect. It should be reworded.

Page 7, line 237:
"... leads higher electron temperature and more intensity ..." -> "... leads to higher electron temperature and higher intensity of ..."

Author Response

We want to began by thanking you for the detailed comments. These comments are all valuable and very helpful for revising and improving our paper. We have studied comments carefully and have made correction which we hope meet with approval. All the points raised by the reviewer are addressed in the paper, as summarized below:

Comment 1: Page 1, line 10, 39:
"invested" -> "investigated"

Response: We are very sorry for our incorrect writing, according to the reviewer’s comment, we have corrected the word.

Comment 2: Page 1, line 11:
"Theory tin ..." -> "Theoretical tin ..."

Response: According to the reviewer’s comment, we have corrected the word.

Comment 3: Page 1, line 15-16:
The expression "With the EUV CE increases from 0.5% to 1.1%." is not a complete English sentence. Also, "EUV" and "CE" should be spelled out at their first appearance.

Response: According to the reviewer’s comment, we have corrected the sentence to "And the Extreme Ultraviolet (EUV) emission conversion efficiency (CE) is increasing from 0.5% to 1.1%"

Comment 4: Page 1, line 22-23: The authors describe "At present, the most popular photolithography method is 13.5 nm extreme ultraviolet lithography.". However, the reviewer thinks that the EUV lithography be still under development and not the most popular.

Response: I’m sorry to make a misunderstanding here. And we have corrected the sentence to "At present, the 13.5 nm extreme ultraviolet lithography (EUVL) is one of the most potential photolithography method".

Comment 5: Page 1, line 23:
"choosing" -> "chosen"

Response: We have corrected the word in the revised paper.

Comment 6: Page 2, line 61:
"The laser beam is perpendicularly ..." -> "The laser beam is injected perpendicularly ..."

Response: We have corrected the sentence in the revised paper.

Comment 7: Page 2, line 66:
The sentence "According the experiments, five shots the spectra are not found to change significantly." is hard to understand. English should be improved.

Response: It is really true as Reviewer’ suggesting. Here we mean that the spectra averaged by five shots will not change significantly under the same discharge condition. And we delete this sentence in the revised paper.

Comment 8: Page 3, Figure 2:
The letters for the insets are too small to read.

Response: Thank you very much for your opinion. We extract the inserted figures as independent figures, which are shown in Figure 2 (b) and (d).

Comment 9: Page 3, Figure 2:
The color for the X-ray signal does not match with the text in the caption.

Response: Thank you for your careful work. We have modified the color of the X-ray signals in the revised paper.

Comment 10: Page 3, line 105:
"show" -> "shows"

Response: We have corrected the word in the revised paper.

Comment 11: Page 3, line 111:
"24J power energy" -> "24J energy"

Response: We have corrected the sentence in the revised paper.

Comment 12: Page 3, line 111:
The authors describe "As shown in Figure 2(a), no discharge occurs at dt smaller than 0.68 μs, ...". However, Figure 2(a) does not show such a fact.

Response: Thank you very much for your valuable advice. In fact, many experiments have been done by increasing T2, which is shown in Figure 1(b). When T2 is large enough, the pre-ionized plasma will be generated after that the high voltage is loaded between the two electrodes. In this way, the high voltage will be maintained, which is similar to Figure 2(a). Defined the time interval between the falling edge of the voltage and the onset of the current as dt’, which is shown in figure 2(a). And the experimental results show that dt’ is increasing with the increasing of T2. However, the laser-current delay dt (~10% of laser pulse to 10% of current) is almost 0.68 μs under different T2.

Comment 13: Page 3, line 116:
"By comparing with the two ..." -> "By comparing the two ..."

Response: We have corrected the sentence in the revised paper.

Comment 14: Page 4, line 134:
"... used to calculated the averaged wavelength verse ion stages ..." -> "... used to calculate the averaged wavelength versus ion stages ..."

Response: We have corrected the sentence in the revised paper.

Comment 15: Page 4, Figure 3:
The "(a)" and "(b)" have not been denoted in the figures.

Response: I’m sorry to make a misunderstanding here. And we have corrected Figure 3.

Comment 16: Page 4, Figure 3:
Why the average wavelength of 4p-4d transition for ion stage of 5 is irregular?

Response: Thank you for your careful work. We have recalculated this part, and the result is the same as that in figure 3(a). In this paper, the transition arrays are calculated by using Cowan code. In our opinion, the reason may be these:

According the theoretical results, there are only a few lines generated by the Sn5+ 4p-4d transitions. However, there are hundreds of lines generated by the Sn6+ 4p-4d transitions. And much higher ion stage’ 4d-4p transition is complex as well. According to Equation (1), the average wavelength of different transitions and different ion stages is the weighted average of the wavelength and transition probability gA. In practice, the wavelengths generated by Sn6+ 4p-4d transitions range from 12.8-17.4nm, which is much wider than the wavelength generated by Sn5+ 4p-4d transitions. Due to higher gA value for longer wavelength, the averaged wavelength for Sn6+ 4p-4d is longer than Sn5+ 4p-4d transition.

Comment 17: Page 4, Figure 3:
"Theory results ..." -> "Theoretical results ..."

Response: We have corrected the word in the revised paper.

Comment 18: Page 4, Figure 3:
The colors for the 4d-5f and 4p-4d do not match with the text in the caption.

Response: Thank you for your careful work. We have modified the color and shapes for the 4d-5f and 4p-4d transitions in Figure 3.

Comment 19: Page 4, line 145:
"... in the Equation (2) for collisional ..." -> "... in the Equation (2) for rate coefficients of collisional ..."

Response: We have corrected the sentence in the revised paper.

Comment 20: Page 5, line 157:
"show" -> "shows"

Response: We have corrected the word in the revised paper.

Comment 21: Page 5, Figure 4:
"Theory tin ..." -> "Theoretical tin ..."

Response: We have corrected the word in the revised paper.

Comment 22: Page 5, Figure 5:
The origin of the vertical axis should start with zero.

Response: Thank you very much for your opinion. And we have modified figure 5 in the revised paper.

Comment 23: Page 6, line 195:
"Figure 6" -> "Figure 5"

Response: Thank you very much for your opinion. And we have modified the word in the revised paper. 

Comment 24: Page 6, Figure 6:
The origin of the vertical axis should start with zero.

Response: Thank you very much for your opinion. And we have modified figure 5 in the revised paper.

Comment 25: Page 6, Figure 6:
The color for the theoretical spectra does not match with the text in the caption. Also, the letters for the insets are too small to read.

Response: Thank you for your careful work. We have modified the shape of the line for the theoretical spectra. Moreover, the figures are enlarged as well.

Comment 26: Page 6, line 204:
"CE" should be spelled out at their first appearance.

Response: Thank you very much for your opinion. Here the CE refers to conversion efficiency, and I have added it in the revised paper.

Comment 27: Page 6, line 207-208:
"On contrast, ..." -> "In contrast, ..."

Response: We have corrected the word in the revised paper.

Comment 28: Page 6, Figure 7:
The origin of the vertical and horizontal axes should start with zero.

Response: We have modified figure 7 in the revised paper.

Comment 29: Page 6, line 215:
The expression "..., here the plasma is referred to that which can generate 13.5 nm emission." is hard to understand what it means. It should be reworded.

Response: According to the reviewer’s comment, we have modified the sentence to "For the EUV source, the 13.5 nm emission intensity in 2% band is defined by the lifetime and density of the plasma which can generate spectra around 13.5 nm."

Comment 30: Page 7, line 222:
What is "the ratio"?

Response: Here "the ratio" is referred to the abundance ratio between the density of Sn8+ - Sn13+ ions and the total density of the plasma. And we have corrected the sentence in the revised paper.

Comment 31: Page 7, line 224:
"... should increases." -> "... should increase."

Response: According to the reviewer’s comment, we have corrected the word.

Comment 32: Page 7, line 225:
The expression "... makes the pre-ionized plasma cannot be pinched ..." is grammatically incorrect. It should be reworded.

Response: According to the reviewer’s comment, we corrected the sentence as follows: “In other words, limited current leads to limited Z-pinch Lorentz force. When the Z-pinch Lorentz force is limited, the pre-ionized plasma cannot be pinched acutely as well. As a result, the electron temperature is not high enough to get more enough Sn8+ - Sn13+ ions”

Comment 33: Page 7, line 237:
"... leads higher electron temperature and more intensity ..." -> "... leads to higher electron temperature and higher intensity of ..."

Response: According to the reviewer’s comment, we have corrected the sentence.

We tried our best to improve the manuscript and made some changes in the manuscript. These changes will not influence the content and framework of the paper. And here we did not list changes but marked in revised paper.

We appreciate for your warm work earnestly, and hope that the correction will meet with approval.

Once again, thank you very much for your comments and suggestions.

Reviewer 2 Report

The same authors have published recently on the Simmetry journal a very similar paper, where they discuss "Effect of Time Delay on Laser-Triggered Discharge Plasma for a Beyond EUV Source".
The paper submitted looks very similar in the architecture, experimental results and discussions.
The authors don't mention the previous paper neither discuss the differences. What is the novelty of this paper ?
It seems the authors are presenting just a different anode with the same experimental setup. Why a new paper ? This should have been just an additional paragraph on the previous paper.
Also the theoretical analysis is exactly the same of the previous paper. Figures are also very similar.
The paper in the present form cannot be published. There is not evidence of any real novelty with respect to the paper already published on Simmetry.

Author Response

We want to began by thanking you for the detailed comments. These comments are all valuable and very helpful for revising and improving our paper. We have studied comments carefully and have made correction which we hope meet with approval. All the points raised by the reviewer are addressed in the paper, as summarized below:

Comment: The same authors have published recently on the Simmetry journal a very similar paper, where they discuss "Effect of Time Delay on Laser-Triggered Discharge Plasma for a Beyond EUV Source".

The paper submitted looks very similar in the architecture, experimental results and discussions.

The authors don't mention the previous paper neither discuss the differences. What is the novelty of this paper ?

It seems the authors are presenting just a different anode with the same experimental setup. Why a new paper ? This should have been just an additional paragraph on the previous paper.

Also the theoretical analysis is exactly the same of the previous paper. Figures are also very similar.

The paper in the present form cannot be published. There is not evidence of any real novelty with respect to the paper already published on Simmetry.

Response: 

The paper submitted looks similar in the architecture, experimental results and discussions. However, there are many differences for the two papers, which are as summarized below:

As mentioned in the introduction part of this paper, the researches on the EUV source based on the LDP mechanism of Sn medium, mainly studies the dynamic evolution of the plasma under the initial stage, the Z-pinch process and the post-discharge stage by using various types of CCD to measure the plasma images. Moreover, the 13.5nm emission energy in 2% band and conversion efficiency (CE) under different working conditions are measured by energy monitor. And the 13.5nm emission energy and CE are used to research the influence of laser parameters, discharge voltage and current on the EUV source. However, there are few studies on the spectra of Sn LDP EUV source, and there are almost no reports on the influence of laser current delay on the spectrum.At present, there are many works on the spectra of Sn LPP source under different working conditions. It is found that the laser-triggered discharge plasma is optical thin by comparing with the laser produced plasma. And the self-absorption effect is weak as well. In this way, there are some differences of the spectra between the two types of EUV source. Therefore, analysis of the influence of different laser-current delays on the Sn LDP spectra becomes important. And it can also be used for the research of Sn LPP EUV source.

In this experiment, the X-ray diode, which is covered by a 500 nm thick zirconium metal filter, is used to measure the optical signal ranging from 6.5 nm to 16.8 nm under different laser-delays. The results show that when the delay is long enough, there may be multiple pinches for the reason that the plasma density decreases with the laser-current delay. Due to the existence of multiple pinches effect, the plasma will be longer in the high temperature condition. In this way, the mean temperature of the plasma will be got. And the 13.5nm radiation intensity increases as well. However, in the previous paper, the 6.7nm radiation discussed in Gd was absorbed seriously by Zr, and the X-ray signal cannot be detected by this experimental apparatus.

In order to increase the impact for application, we measure the 13.5nm (2% band) emission energy and conversion efficiency under different laser-current delays by a Bruker EUV energy monitor E-Mon. And the results show that, with fixed discharge energy but higher laser-current delay, lower vapor velocity leads higher electron temperature and higher intensity 13.5 nm emission in 2% band ,which increases about 120%. And this experiment has not been done in the previous article.

The theoretical results of Sn atomic parameters are quite different from those of Gd atomic parameters, especially the relationship between the average wavelength and the degree of ionization. For Gd elements, the 4-4 transition is the main source of the radiation near 6.7nm. With the increase of the degree of ionization, the average wavelength increases first and then decreases. When the degree of ionization is 17 to 19, the average wavelength reaches the minimum value. For Sn element, as shown in figure 3 (a), the radiation near 13.5nm mainly comes from Sn8+- Sn13+ 4d-4f and 4p-4d transitions. With the increase of ionization degree, the average wavelength basically decreases. These differences lead to great differences in the spectral simulation and theoretical comparison of Gd and Sn with the experimental results.

In the previous paper, the main research is based on 6.7nm Gd LDP BEUV source, and the range of the wavelength is 6-8 nm. In this paper, a 13.5nm EUV source based on Sn is discussed, and the range of the wavelength is 12-16 nm. Due to the similarity of research methods, there are some articles put the research results of EUV and BEUV source in one article, such as (O'Sullivan g. et al Journal of Physics B: Atomic, molecular and optical physics. 2015, 48, 144025), but these articles are mainly review. On the other hand, for example, in the filed of LPP source, the types of laser (such as CO2and Nd : YAG laser system), width of the laser pulse , types of targets (pure target and oxide, just like Sn and SnO2), the influence of single and double laser pulse and so on are studied in experiments for EUV and BEUV source. Due to different application background, most of the articles are studied independently. In this way, in our opinion, we have not add this paper as an additional paragraph to the previous paper.

When we wrote this paper, we thought that these two paper are in different application backgrounds. In this way, we have not mention the previous paper. In the revised paper, according to the comments, the referenceof the previous article is inserted in the revised

Moreover, we tried our best to improve the manuscript and made some changes in the manuscript. These changes will not influence the content and framework of the paper. And here we did not list changes but marked in revised paper.

We appreciate for your warm work earnestly, and hope that the correction will meet with approval.

Once again, thank you very much for your comments and suggestions.

Reviewer 3 Report

Xiaolong Deng, et al. investigated the influence of the pre-ionized plasma on the dynamics of a tine laser-triggered discharge-plasma, particularly the laser-current delay on the extreme ultraviolet emission. The authors compared the typical wave-forms of current, voltage, laser signal, as well as the X-ray signal. The simulated tin spectra were also compared with the experimental results. They found that the extension of the laser-current delay time can increase the steady-state time of plasma under a high temperature. It is a good work about the plasma physics.

I have the following recommendations for a minor revision.

I recommend the authors improve the English wring through the whole manuscript. If possible, please provide the photo corresponding to the schematic illustration of the experimental setup. In Fig. 6, a and c have shown a good agree between the experimental data and the simulation data. The Fig. 6b is clearly a different case. I recommend the authors explain more or discuss more about such difference. There are clear error bars in Fig. 7. I recommend the authors also provide the error bars in all figures involving data, if possible.

Author Response

Revision-author’s response

Referee #3:

We want to began by thanking Referee #3 for the detailed comments. These comments are all valuable and very helpful for revising and improving our paper. We have studied comments carefully and have made correction which we hope meet with approval. All the points raised by the reviewer are addressed in the paper, as summarized below:

Comment 1: I recommend the authors improve the English wring through the whole manuscript. 

Response: It is really true as Reviewer’s suggested, we have modified the English writing through the whole manuscript, which are marked in the revised paper.

Comment 2: If possible, please provide the photo corresponding to the schematic illustration of the experimental setup. 

Response: Thank you for your valuable advice. And we have added the photo of the EUV source into figure 1(b).

Figure 1(b). Photograph of the experimental setup from the side of the grazing incidence spectrometer of the vacuum chamber.

Comment 3: In Fig. 6, a and c have shown a good agree between the experimental data and the simulation data. The Fig. 6b is clearly a different case. I recommend the authors explain more or discuss more about such difference. 

Response: Thank you for your valuable advice. As mentioned in the paper, the figure 6 represents the comparisons of the same spectrum and the simulated spectrum under different electron temperatures. According to figure 3(a) and figure 6(a), the spectra between 13 nm an 14 nm are mainly originated from Sn10+ and Sn12+, and these ions mainly exist under the electron temperature 32eV. In this way, the experimental data and the simulation data can agree well. And the condition for figure 6(c) is for the same reason. However, It can be seen from the figure 3(a) that the origin of spectra in the range of 14-15nm is particularly complex, which are mainly coming from Sn6+ ~ Sn13+ ions. And these ions can exist in a very large temperature range. At the same time, the measured spectrum is a time integral spectrum, which corresponds to the overall situation of plasma radiation spectrum at different temperatures during the whole Z-pinch process. This means that the spectra in this band can not match the fitting spectra at a certain temperature, which is also the main reason why the theoretical and experimental spectral in figure 6(b) can’t fit well.

And we have added these sentences in the revised paper.

Comment 4: There are clear error bars in Fig. 7. I recommend the authors also provide the error bars in all figures involving data, if possible.

Response: 

It is really true as Reviewer’ suggesting. 

There are seven figures in this paper.

Figure 1 is the schematic of experimental setup.

Figure 2 are the typical wave-forms for the voltage, current, IRD diode X-ray signal, and the laser signal under two laser-current delays. These wave-forms are measured by various detectors, and they are typical wave-forms. In this way, we think that it cannot be provided the error bars as well.

Figure 3 are the theoretical results of averaged wavelength and ion fraction. In figure 3(a), we have provided the error bars, which reflects the distribution of different wavelengths calculated under the certain ionization degree and the certain transition condition. Figure 3(b) is the fractional ion stage distributions and average ions stage in an LTD tin plasma. According to the theoretical model, the fractional ion stage distributions and average ions stage are certain values at a certain electron temperature and density. In this way, we think that it cannot be provided the error bars as well.

Figure 4 show the theoretical tin spectra as a function of temperature and wavelength. According to the theoretical result, the spontaneous transition probability A,  statistical weight g, wavelength et al. parameters are the certain values. As a result, the theoretical tin spectra is certain at a certain temperature. In this way, we think that it cannot be provided the error bars as well.

Figure 5 and 6 are typical EUV spectra, and they cannot be provided error bars as well.

In this way, we can provide error bars in figure 3 (a) and figure 7.

We tried our best to improve the manuscript and made some changes in the manuscript. These changes will not influence the content and framework of the paper. And here we did not list changes but marked in revised paper.

We appreciate for your warm work earnestly, and hope that the correction will meet with approval.

Once again, thank you very much for your comments and suggestions.

Round 2

Reviewer 2 Report

I would like to thank the authors for the answer to the main questions regarding the partial overlapping with a recent paper from the same group. They provided sufficient information to clarify the difference between the two papers and to justify the publication of this new one.

The results are clearly presented and discussed. In the present form, the paper can be published.